# The Molecular Chaperone TCP1 Affects Carcinogenicity and Is a Potential Therapeutic Target for Acute Myeloid Leukemia

**DOI:** 10.3390/pharmaceutics17050557

**Published:** 2025-04-24

**Authors:** Yong Wu, Guihui Tu, Yuxia Yuan, Jingwen Liu, Qingna Jiang, Yang Liu, Qiurong Wu, Lixian Wu, Yuanzhong Chen

**Affiliations:** 1Fujian Provincial Key Laboratory on Hematology, Fujian Institute of Hematology, Fujian Medical University (FMU), Union Hospital, Fuzhou 350122, China; wuyong9195@126.com; 2Department of Pharmacology, School of Pharmacy, Fujian Medical University (FMU), Fuzhou 350122, China; tghinfuzhou@163.com (G.T.); yyx19960829@163.com (Y.Y.); 2023024@fjtcm.edu.cn (J.L.); qingna@fjmu.edu.cn (Q.J.); wuqiurong@outlook.com (Q.W.); 3Department of Pharmacochemistry, School of Pharmacy, Fujian Medical University (FMU), Fuzhou 350122, China; liuyang966@163.com

**Keywords:** acute myeloid leukemia, TCP1, miR-340-5p, FTY720, cell cycle arrest, chaperone function

## Abstract

**Background/Objectives:** Acute myeloid leukemia (AML) is an aggressive malignancy marked by high relapse rates and molecular heterogeneity, necessitating the identification of novel therapeutic targets. T-complex protein 1 (TCP1), a chaperonin implicated in protein folding, remains underexplored in AML pathogenesis. This study investigates the functional role of TCP1 in AML progression and evaluates its therapeutic potential. **Methods:** Using successive generations of xenografted tumor models, we systematically assessed the correlation between TCP1 expression and AML tumorigenicity. Functional consequences of TCP1 silence were evaluated through in vitro proliferation assays and in vivo tumor growth monitoring. Two distinct inhibitory strategies were employed: miR-340-5p-mediated transcriptional silencing and FTY720-induced disruption of TCP1 chaperone activity. Mechanistic insights were derived from ubiquitin–proteasome pathway analysis, cell cycle profiling, and apoptosis assays. **Results:** High TCP1 expression correlated strongly with enhanced AML tumorigenicity. Knockdown of TCP1 significantly inhibited AML cell growth and induced degradation of AML1-ETO and PLK1 proteins through the ubiquitin–proteasome pathway. miR-340-5p effectively silenced TCP1 expression, exhibiting an inverse correlation with TCP1 levels. FTY720 disrupted TCP1′s chaperone function, leading to cell cycle arrest, apoptosis, and reduced xenograft tumor growth in murine models. **Conclusion:** Our findings establish TCP1 as a promising therapeutic target for AML. Both miR-340-5p and FTY720 demonstrate potent anti-leukemic effects by suppressing TCP1 activity, highlighting their potential as novel strategies to inhibit AML proliferation and improve therapeutic outcomes.

## 1. Introduction

Acute myeloid leukemia (AML) stands as a formidable adversary in the landscape of oncology, characterized by its rapid cell proliferation, invasive nature, and high mortality rates. Despite recent advances in therapeutic strategies, including the approval of novel drugs by the FDA, the heterogeneity of AML poses a significant challenge, with patients experiencing high recurrence rates and a pressing need for more effective treatments [1,2,3].

Our investigation delves into the role of T-complex protein 1 (TCP1), a subunit of the chaperonin-containing T complex (CCT), which is integral to the folding of newly synthesized proteins and the maintenance of cellular protein homeostasis. The significance of TCP1 extends beyond its chaperone function, with its potential involvement in the pathogenesis of AML. CCT is essential for folding actin, a cytoskeletal protein [4]. It dismantles the mitotic checkpoint complex (MCC) to ensure proper chromosome–spindle connections [5] and triggers the anaphase-promoting complex/cyclosome (APC/C) by liberating CDC20, leading to MCC disintegration [6]. CCT influences G1/S phase proteins cyclin E and Cdk2 [7,8], promoting cyclin E maturation and impacting the cell cycle if absent [7,9]. It also aids in folding of STAT3 [10] and huntingtin (Htt) [11] and is linked to various proteins and pathways, including MYC, CCDN1 [12], PP2A [13], Wnt7b/β-catenin, P53 [14], AKT/mTOR [4], and the AML1-ETO fusion protein in M2-type AML [15,16]. Furthermore, our group’s preliminary study found that targeting the TCP1/AKT/mTOR signaling pathway could overcome drug resistance in AML cells [4]. This study observed that TCP1 levels increased in successive generations of xenografted tumors, correlating with the enhanced tumorigenicity of AML cells. This correlation suggests a pivotal role for TCP1 in the modulation of AML cell and tumor growth, particularly through its influence on proliferation signaling pathways. However, it has not been reported whether TCP1 can be an effective target for AML treatment.

In our study, we used two strategies to analyze whether targeted TCP1 can treat AML. MicroRNAs (miRNAs) are transcribed in the nucleus by RNA polymerase II. They find and bind to specific mRNAs by matching their bases, which can break down the mRNA or stop its translation. One mRNA can be targeted by many miRNAs. Therefore, first, we explored whether there are miRNAs that can specifically silence or inhibit the expression of TCP1 by regulating TCP1 at the mRNA level. We also screened and validated that miR-340-5p can specifically target the TCP1, inhibiting AML cell proliferation.

Second, we used SybylX2.0 to search 2532 FDA-approved drugs for those that interact well with the TCP1 protein. We found about 20 potential compounds and, through cell activity tests, narrowed it down to 5 drugs with good activity. The best one was Fingolimod (FTY720), which is normally used to treat multiple sclerosis by blocking a receptor called S1P. Recently, it has been shown to have anti-cancer effects in various cancers [17,18,19,20,21,22,23,24]. In AML, it can cause AML cells to die by exposing phosphatidylserine (PS) [23]. FTY720 also reduces cancer protein AML1-ETO, leading to cell cycle arrest and cell death through increased pro-apoptotic molecules and reactivation of protein phosphatase 2A (PP2A) [25,26]. However, it has not been reported whether FTY720 can inhibit AML cell proliferation by affecting the function of TCP1.

Here, we explored the importance of TCP1 in the progression of AML and found that AML cells lacking TCP1 did not survive, verifying that TCP1 may be a potential therapeutic target for AML. We further explored the drugs and miRNAs that can target and inhibit TCP1 and their possible mechanisms to propose possible solutions by targeting TCP1 for treating AML.

## 2. Materials and Methods

### 2.1. Clinical Samples

The clinical samples of healthy donors and AML patients were from the hematology department of Fujian Medical University Union Hospital. A total of 84 peripheral blood samples from donors were collected, including 24 after the first visit, 45 after treatment, and 15 samples from healthy donors. All donors signed the informed consent form under the supervision of the Institutional Review Board of Fujian Medical University Union Hospital (Fuzhou, China; Approval No: 2017KY091).

### 2.2. Cell Lines

HL-60, KG1α, and Kasumi-1 cells were obtained from the Shanghai Institute of Biochemistry and Cell Biology, Shanghai, China. The HL-60-G4 cell line was collected and cultured from the fourth-generation tumor formed by the HL-60 cells in nude mice. All the cell lines were cultured in RPMI-1640 (Cytiva, Marlborough, MA, USA) with 10% FBS (PAN-Biotech, Aidenbach, Germany), 100 U·mL^−1^ penicillin, and 100 μg·mL^−1^ streptomycin (NCM Biotech, Bucheon-si, Republic of Korea) in a humidified incubator in 5% CO_2_ at 37 °C. All cell lines were authenticated by short tandem repeat (STR).

### 2.3. Compounds and Reagents

The following compounds were used: FTY720 (GlpBio, GC14807), NSC348884 (GlpBio, GC16151), Cediranib (AZD2171) (GlpBio, GC16421), Foretinib (GlpBio, GC15735), Lipitor (Selleck, 134523-03-8), hsa-miR-340-5p inhibitor (MCE, HY-RI00707).

The following antibodies were used: anti-TCP1 polyclonal antibody (Proteintech, 10320-1-AP); anti-cleaved-caspase-3 (Wanleibio, WL02117); anti-CDC20 Rabbit pAb (Abclonal, A15656); anti-PKMYT1 Rabbit pAb (Abclonal, A20525); anti-GAPDH monoclonal antibody (ImmunoWay, YM3029); anti-AML1-ETO (CST, 4334); anti-cleaved-caspase-9 (CST, 52873); anti-cleaved-PARP (CST, 5625); anti-PLK1 (Abcam, ab189139); anti-pan-myc (Abcam, Ab195207); anti-mouse IgG, HRP-conjugated (CST, 7076); and anti-rabbit IgG, HRP-conjugated (CST, 7074).

### 2.4. Different Generations of Xenograft Tumor Models

We obtained 16–20 g, 6-week-old male BALB/c nude mice from Fujian Minhou Wu Experimental Animal Trading Co. Ltd. (Fuzhou, China)and fed them in specific pathogen-free conditions. All animal studies reported here were approved by the Institutional Review Board of Fujian Medical University.

For the establishment of the tumor model and the highly tumorigenic HL-60 cells, 10 nude mice were implanted with HL-60 cells (2 × 10^6^ cells) and allowed tumors to grow for a month before being sacrificed. The tumor tissues were removed and digested, and the cells, named HL-60-G1, were isolated. The cells were implanted subcutaneously into nude mice again to form tumors after 15 passages, and then HL-60-G2 cells were isolated and cultured in the same way. After repeating the procedure twice, four generations of cell lines were obtained.

### 2.5. Proteomic Analysis

Proteomic analysis was conducted as previously described [27]. In short, the proteins derived from different generations of cells were extracted and randomly labelled with Cy3 or Cy5. Subsequently, the samples were mixed and separated with 2D-DIGE, and a Typhoon 9410 scanner combined with Decyder 6.5 software were used to normalize and quantify the protein spots in the gels. At last, the peptides were identified by a Proteomics Analyzer.

### 2.6. Quantitative Real-Time PCR Analysis

The total RNA from the tissue samples of AML patients were extracted with the TRIzol reagent (Invitrogen, 15596026CN, Waltham, MA, USA), and the cDNA was synthesized using HRbioTM III 1st Strand cDNA Synthesis Kit (HeRui, HRF0192, Guangzhou, China). qPCR was performed on a Light Cycler 96 Real-Time PCR system (Roche, Shanghai, China) with HRbioTM qPCR SYBR Green Master Mix (HeRui, HRF0032, Guangzhou, China). Gene relative expression was normalized to that of GAPDH by the 2^−ΔΔCt^ method. The forward and reverse primers are listed as follows:TCP1-Forward: AGCCACGCTATCCAGTCAAC;TCP1-Reverse: GGCTGAAGTCAAGGCAAGCA;hsa-miR-340-5p-Forward: GCCGCGCTTATAAAGCAATGAGACTGA.

### 2.7. Western Blot Analysis

We seeded 2 × 10^6^ HL-60-G4, KG1α and Kasumi-1 cells in each well in 6-well plates and added a specified concentration of FTY720 for 24 h. Then, cells were lysed with RIPA lysis buffer on ice. The protein concentration was determined with a BCA assay kit (DINGGUO, BCA02). The proteins were loaded on a 10% gel and then transferred to PVDF membranes. Next, the bands were blocked with 5% skim milk for 2 h and incubated with the primary antibody overnight at 4 °C. After incubating with the secondary antibody for 2 h at room temperature, the bands were washed with TBST and tested with the Gel Doc XR imaging system (Bio-Rad, Hercules, CA, USA) using an ECL substrate reagent kit (Smart-Lifesciences, H31500, Changzhou, China).

### 2.8. Lentivirus Transfection

For lentivirus transduction of TCP1 and Tet-on system, PCR was conducted to amplify the positive lentiviral vectors containing recombinant human TCP1 gene as well as the Tet-On system obtained from Shanghai Genechem Co. Ltd. (Shanghai, China). As HL-60-G4 cells reached 80% confluence, the cells were centrifuged, counted, and replated in culture dishes and then exposed to lentivirus at multiplicities of infection (MOI) of 50 with 7 μg·mL^−1^ Polybrene (Sigma, St. Louis, MO, USA) to enhance the transduction efficiency.

For knockdown TCP1 in HL-60 cells, we transfected shTCP1 and the negative control group, respectively, into HEK293T cells together with plasmids pVSVG, pREV, and pMD (Genechem, Shanghai, China) and extracted fresh lentivirus to transfect HL-60 or HL-60-G4 cells. After 72 h of transfection, all cells were screened with puromycin (1 µg·mL^−1^) for half of a month. Then, the successfully transfected cells were verified, and the efficiency of transfection and the level of TCP1 knockdown were determined by Western blotting. The optimal sequences of TCP1 short hairpin RNA (shRNA) were as follows:

TCP1-shRNA/F: 5′-CCGGGGTGTACAGGTGGTCATTATTCAAGAGATAATGAC CACCTGTACACCTTTTTTG-3′;

TCP1-shRNA/R: 5′-AATTCAAAAAAGGTGTACAGGTGGTCATTATCTCTTGAA TAATGACCACCTGTACACC-3′.

### 2.9. Protein Profile Analysis

TCP1 knockdown and control cells were generated by transfecting HL-60-G4 cells with lentivirus-mediated TCP1-shRNA or NC-shRNA. After that, total RNA was extracted and purified using an RNeasy Mini kit (Qiagen, Hilden, Germany). RNA quality assessment was carried out by NanoDrop 2000 (Thermo, Waltham, MA, USA) and agarose gel electrophoresis. Then, the differential expression genes (DEGs) between shNC and shTCP1 were identified based on *p* value < 0.05 and fold change >1.2 or <0.833. The PPI network for the DEGs was constructed using the Cytoscape software (Version 3.7.2). The function and pathway enrichment analyses of DEGs were performed using DAVID.

### 2.10. Screen and Selection of the miRNA

For predicting the target microRNA of TCP1, the target microRNA appeared in the databases of TargetScan (https://www.targetscan.org/vert_71/. Accessed on 1 January 2022) and was selected to evaluate the regulatory relationships. In detail, only targets with a cumulative weighted context++ score of 0.90 and above were considered.

### 2.11. Luciferase Reporter Gene Assay

The plasmid bacteria containing the complete and correct 3′-UTR region of the target protein was spread on the solid LB medium containing antibiotics. We picked a single colony to inoculate in a liquid LB medium and incubate overnight at 200 rpm in a shaking box, then inoculated the bacterial solution in a 250 mL conical flask at a ratio of 1:100 and cultured it at 37 °C overnight. The plasmid was extracted using the Plasmid Midi Kit according to the instructions (TIANGEN, DP108).

The 293T cells were deposited into a six-well plate, and the mimic, the plasmid, and the transfection reagent Lipofection 2000 (Invitrogen, 11668-019, Waltham, MA, USA) were added, followed by incubation for 48 h. The luminescence unit detection of dual-luciferase was performed according to the instructions of the Dual-Luciferase Reporter Assay System (Promega, E1910, Madison, WI, USA). Firefly fluorescence value/sea cucumber fluorescence value was used for data processing. All results were standardized and compared with the negative control group (NC).

### 2.12. SYBYL Software Analysis

We divided the TCP1 protein crystal structure from the CCT (PDB id = 4B2T) protein crystal structure downloaded from the Protein Data Bank (PDB) website. Because there was not a precise protein binding pocket in TCP1, we first chose the most possible and suitable sites as the protein binding pocket by the computer-aided software Sybyl-X 2.0 to combine with drugs. We used the Surflex-dock program to screen 2532 FDA-approved active pharmaceutical ingredients (APIs) and TCP1 protein in the FDA-approved drug library provided by Selleck for virtual screening of molecular docking and selected drugs with high docking scores to further research the activity and affinity with TCP1.

### 2.13. MTT Assay

HL-60 (3 × 10^4^ cells/well), KG1α (2×10^4^ cells/well), and Kasumi-1 (2.5 × 10^4^ cells/well) cells were deposited into 96-well plates and incubated with FTY720 from 100 μmol·L^−1^ to 0.390625 μmol·L^−1^. After 48 h, MTT was added to the medium and incubated for 4 h. Then, the 96-well plates were centrifuged to discard the supernatant, and 200 μL DMSO was added to dissolve the crystallization. Finally, the *OD* value at 570 nm of each well was measured by a full-wavelength microplate reader (Thermo). The GraphPad Prism software (Version 8.0.1) was used to draw the growth curve.

### 2.14. Surface Plasmon Resonance Analysis

CM5 chip and sodium acetate buffer (PH 4.5) were prepared and placed in the Biacore T200 equipment, and selected Tools-prime was used to change the buffer system. The CM5 chip was pre-activated by EDC/NHS, and 20 μg·mL^−1^ TCP1 (Abnova, H00006950-P01, Taipei, Taiwan) solution was prepared by dissolving in ultra-pure water to fix the TCP1 protein on the chip by amino-coupling assay. Then, the chip was blocked by ethanolamine. A solution of FTY720, NSC348884, Cediranib (AZD2171), Foretinib, and Lipitor was prepared in different concentrations—100, 50, 25, 12.5, 6.25, 3.125, 1.5625 μmol·L^−1^ and injected into the channel to detect if there was binding and dissociation between TCP1 and the compounds. The KD value was calculated to measure the degree of interaction between TCP1 and FTY720.

### 2.15. Cell Colony Assay

0.7% and 1.2% agarose solutions were prepared and stored at 42 °C to prevent solidification; the mixed agarose solution had twice the concentration of the 1640 medium at a rate of 1:1. 1 mL of the 1.2% agarose mixed solution was then added to each well in the 12-well plate to solidify at room temperature and placed in the cell incubator. AML cell lines were treated with FTY720 for 4 h and mixed with the 0.7% agarose mixed solution. The cell hybrid suspension was added over the solidified 1.2% agarose mixed solution to let it clot at room temperature. 200 μL of the 1640 medium was then added on the 3rd day and cultured for it 14 d to check the cell colonies number. Images were taken with an optical microscope to compare the number of colony units.

### 2.16. Apoptosis Analysis

AML cell lines were treated with FTY720 for 4 h and washed with the PBS solution twice. Then, the cells were incubated with Annexin V-APC and 7-AAD for 20 min to analyze on the BD FACSCanto (TM) II (BD).

### 2.17. Cell Cycle Analysis

AML cell lines were treated with FTY720 for 24 h and washed with the PBS solution twice. Then, 250 μL of PBS was resuspended, followed by 750 μL of absolute ethanol drop by drop with shocking, and the solution was stored at −20 °C for 24 h to fix the cells. On the second day, they were cleared away, and the ethanol solution was washed with the PBS solution twice. Finally, the cells were resuspended in Flow Cytometry Staining Buffer (Proteintech, PF00018, Wuhan, China) and incubated in the dark for 30 min to analyze on the BD FACSCanto (TM) II (BD).

### 2.18. Co-Immunoprecipitation Analysis

6 × 10^7^ HL-60-G4 and 3 × 10^7^ Kasumi-1 cells were seeded and added to FTY720 at 30 μmol·L^−1^ or 40 μmol·L^−1^ respectively, for 24 h. The protein lysate was divided into an input group, an IgG group, and an IP group. 20 μL of pre-washed Protein A/G Magnetic Beads (MCE, HY-K0202) was added to combine the non-specific substance and the collected protein lysate. 1 μg of the TCP1 antibody was added into the IP group, and 1 μg of Normal Rabbit IgG (CST, 2729) was added into the IgG group at 4 °C for 12 h. On the second day, 20 μL of pre-washed Protein A/G Magnetic Beads was added to the TCP1/PLK1, TCP1/AML1-ETO, or TCP1/CDC20 protein complex and stored at 4 °C for 6 h before collecting the beads. The protein connecting to the beads was dissociated by heating at 98 °C for 10 min with 1× the loading buffer. Finally, the protein level was detected by Western blotting.

### 2.19. The Effect of FTY720 on Xenograft AML Tumors

Every mouse was injected with 1 × 10^7^ HL-60-G4 cells into the right armpit subcutaneously, and we waited for the volume of the tumor to reach about 100 mm^3^ before beginning intraperitoneal administration. The mice were randomly assigned to a control group, a 5 mg kg^1^ FTY720 group, or a 10 mg kg^−1^ FTY720 group. Every two days, we intraperitoneally injected the FTY720 solution or normal saline and detected the body weight and tumor volume of mice until the tumor volume reached 2000 mm^3^.

### 2.20. Statistical Analysis

For all the statistical analyses involved in this study, the significance was set as *p* < 0.05 (* *p* < 0.05, ** *p* < 0.01, *** *p* < 0.001). All results in this work are presented as the means ± standard errors and repeated three times. Student’s *t*-test was used to compare two groups of independent samples, and one-way ANOVA analysis was used to evaluate the statistical significance of three or more separate groups.

## 3. Results

### 3.1. Elevated TCP1 Expression in Successive Xenograft AML Tumor Generations Correlates with HL-60 Cell Tumorigenicity

Previous studies have shown that successive transplantation of HL-60 cells in nude mice could enhance the tumorigenicity in vivo [28]. Consistent with this, our study constructed four generations of HL-60 cell lines and found that tumorigenicity increased as the generations increased (Figure 1A). To explain this finding, 2D-DIGE combined with mass spectrometry was performed on different generations of HL-60 lysates (Figure 1B). We observed 2153 spots corresponding to matched proteins on the gels, of which 33 spots presented significant upregulation in highly tumorigenic cells, and 54 spots were downregulated (Figure 1C). Of these, one of the differentially expressed proteins, No. 71, was identified as TCP1 (Figure 1D). To further validate the differential expression of TCP1, we examined the TCP1 in tumor tissues from different generations of HL-60 cells by IHC staining. As shown in Figure 1E, the TCP1 level increased with increasing generations, indicating that it may be closely linked to the development of AML. To test our hypothesis, we used RT-qPCR and Western blotting to examine the tissue samples from AML patients and various leukemia cells. We found that the level of TCP1 was significantly higher in the recurrent and untreated marrow samples of AML than in the normal and remission samples (Figure 1F,G). Likewise, there was also a higher level of TCP1 in various leukemia cells compared to the normal bone marrow (Figure 1H). Furthermore, Kaplan–Meier curves demonstrated that patients with low TCP1 expression experienced prolonged survival (Appendix A). Based on these results, we speculated that TCP1 is highly expressed in AML cells, which may contribute to the tumorigenicity of AML.

### 3.2. TCP1 May Regulate Multiple Signaling Pathways Related to AML Cell Survival

To further explore the importance of TCP1 in AML cells, we established a stable TCP1-knockdown HL-60-G4 cell model via lentiviral transduction. The efficiency of TCP1 depletion was systematically validated through Western blotting (Appendix A). Subsequently, both control cells (wild-type HL-60-G4) and TCP1-knockdown cells were subjected to isobaric tags for relative and absolute quantitation (iTRAQ)-based comparative proteomics. This approach enabled the identification of potential TCP1-associated proteins and downstream molecular pathways affected by TCP1 depletion. We chose differentially expressed proteins (DEPs, fold change ˃1.2 or <0.833, *p* value < 0.05) to calculate the degree value, which represents the importance of the proteins in the network. A total of 140 DEPs with a degree value ˃ 10 were regarded as hub proteins, and PPI, GO, and KEGG pathway enrichment analyses were used to detect the cellular biological processes involved in 140 hub proteins. In the PPI network, we showed the interaction of the hub proteins, where proteins in the circle are related to TCP1 directly, and the other part of proteins on the right may be regulated by TCP1 indirectly. The color of the proteins is darker if the degree value is more considerable, which means that the proteins are connected with and may regulate other proteins (Appendix A). In the KEGG analysis, we selected the first 20 pathways from 160 KEGG terms and found that the pathways were mainly focused on thermogenesis, oxidative phosphorylation, PI3K-Akt signaling pathway, cellular senescence, and Th17 cell differentiation (Appendix A). In the three parts of the GO analysis, there were 117 GO terms with 54 BP terms, 50 CC terms, and 13 MF terms. The top 10 terms also involved the PI3K-Akt signaling pathway, oxidative phosphorylation, and mitochondria (Appendix A). These pathways could affect the cell cycle, apoptosis, and proliferation.

### 3.3. Knocking Down TCP1 Leads to Effective Inhibition of AML Cell Proliferation In Vitro and In Vivo

To evaluate the therapeutic potential of TCP1 as a molecular target in AML, we investigated the effects of TCP1 knockdown on HL-60-G4 cell growth using colony formation assays. As demonstrated in Figure 2A,B, genetic depletion of TCP1 resulted in a marked reduction in cellular proliferation relative to the scrambled group. Cell cycle analysis further revealed a significant accumulation of TCP1-deficient cells at the G2/M phase (Figure 2C,D). These findings collectively indicate that the knockdown of TCP1 could inhibit the proliferation of AML cells by blocking the cell cycle. Furthermore, we used Tet-on technology to construct HL-60-G4-tet-shTCP1 cells that can knock down TCP1 by adding doxycycline (Dox+) and used this cell line to construct a xenograft tumor model in nude mice. Compared to the control group (Dox−), Dox+ significantly induced TCP1 knockdown in tumor tissues and reduced tumor volume (Figure 2E–G), which further verified that targeting TCP1 could be a potential treatment option for AML.

### 3.4. TCP1 Was Able to Maintain the Stability of the PLK1 and AML1-ETO Proteins Through Its Chaperone Function

To further elucidate the potential mechanism of TCP1-mediated AML cell survival, we analyzed its regulatory effects on PLK1 (a cell cycle regulator) and AML1-ETO (a leukemia-associated fusion protein) using AML cell lines with constitutive TCP1 knockdown (HL-60-G4-shTCP1 and Kasumi-1-shTCP1). It was obvious that the knockdown of TCP1 significantly reduced the protein levels of both PLK1 and AML1-ETO in HL-60-G4 and Kasumi-1 cells (Figure 3A), indicating that TCP1 may critically regulate the stability or expression of these factors. We then used the protein synthesis inhibitor cycloheximide (CHX) and proteasome inhibitor MG132 to explore the degradation mechanism of the PLK1 and AML1-ETO proteins. We found that TCP1 knockdown results in faster protein degradation than the control cells after CHX treatment. However, when MG132 was added to the system, the protein degradation was significantly inhibited in comparison to CHX alone, suggesting that TCP1 may influence the protein stability of PLK1 and AML1-ETO through the ubiquitin–proteasome system rather than affecting the new synthesis of proteins (Figure 3B,C). Successively, using Co-IP assays, we found that TCP1 knockdown decreased the pull-down levels of PLK1 and AML1-ETO (Figure 3D). Therefore, suppressing the expression of TCP1 may inhibit the chaperone function of TCP1, thereby reducing the formation of TCP1/PLK1 and TCP1/AML-ETO complexes and ultimately leading to the degradation of PLK1 and AML1-ETO through the proteasome pathway.

### 3.5. MiR-340-5p Can Inhibit the Expression of TCP1, Thereby Blocking AML Cell Proliferation

To analyze the therapeutic effect of targeting TCP1, we used microRNAs (miRNAs) to block the mRNA expression of TCP1 in AML cells. We selected six different sequences of miRNAs from the TargetScanHuman website by the predicted score to perform the dual-luciferase reporter system. Of these, miR-340-5p was found to reduce the most dual-luciferase activity, which meant miR-340-5p was the optimal miRNA for regulating TCP1 mRNA **(**Figure 4A). Next, we identified the potential binding sites between miR-340-5p and TCP1, and the mutated sequence of TCP1 was also displayed (Figure 4B). The effect of miR-340-5p on TCP1 translation was tested by a dual-luciferase reporter system using a luciferase reporter plasmid constructed with the wild-type 3′-UTR of TCP1 or a mutant. We found that the luciferase activity of 293T cells transfected with the wild-type 3′-UTR of TCP1 was reduced by miR-340-5p, but not the mutant. To validate our hypothesis, we analyzed miR-340-5p expression in HL-60-G4 cells (a subline exhibiting elevated TCP1 levels compared to parental HL-60 cells). miR-340-5p expression was significantly downregulated in HL-60-G4 cells (Figure 4C). To further explain its regulatory effect on TCP1, we performed lentivirus-mediated miR-340-5p overexpression and assessed TCP1 protein levels via Western blotting. Overexpression of miR-340-5p substantially suppressed TCP1 protein level and markedly inhibited HL-60-G4 cell proliferation (Figure 4D,E); however, adding a miR-340-5p inhibitor effectively reversed the proliferation inhibition (Figure 4F). All these findings prove that miR-340-5p could negatively regulate the TCP1 to inhibit AML cell proliferation.

### 3.6. FTY720 Has an Affinity for TCP1 and Can Inhibit AML Cell Proliferation

Since we found that TCP1 is a potential target for AML therapy, in the next step, we wanted to determine whether there is a chemical drug that could specifically target TCP1 to treat AML. We used the Surflex-dock program in Sybyl-X 2.0 to perform molecular docking virtual screening on the TCP1 subunit protein and 2532 FDA-approved APIs in the FDA-approved Drug Library provided by Selleck. We obtained 24 hit compounds and determined 5 drugs with better cell activity through the MTT experiment (Appendix A). FTY720 was screened out to have the best affinity *KD* value by the SPR (Biacore) method (Figure 5A). Three hydrogen bonds formed between the chemical structure of FTY720 and the TCP1 protein structure with a score of 8.83, which indicated good binding affinity (Figure 5B). In the Biacore result shown in Figure 5C, FTY720 slowly bound and dissociated from TCP1, and the *KD* value was 6.769 × 10^−9^ M. Moreover, the binding level of FTY720 increased as the concentration of FTY720 increased. These findings indicated that FTY720 showed great binding affinity to TCP1, and perhaps FTY720 could affect the proliferation of AML cells by inhibiting TCP1 function. As shown in Figure 5D, AML cell lines were treated with FTY720 halved from 100 μmol L^−1^ to 0.78125 μmol L^−1^ for 48 h. The percentage of viable cells decreased as the FTY720 concentration increased. These findings indicated that AML cells were sensitive to FTY720.

As shown in Figure 5E–F, HL-60-G4 cells had higher protein levels of TCP1 than HL-60 cells, and HL-60-shTCP1 cells had the lowest level of TCP1 protein. Then, we compared the effect of FTY720 on these cells and found that HL-60-G4 cells were the most sensitive to FTY720, and the IC50 was lower than that of HL-60 cells; HL-60-shTCP1 cells had the highest IC50 but could barely proliferate (Figure 5G). In the cell colony forming experiment (Figure 5H,I), HL-60-G4 cells formed significantly more colony units than HL-60 cells. The cell colony units of both cells decreased as the FTY720 concentration increased. The colony of AML cells could barely be formed after treatment with 20 μmol L^−1^ FTY720. These findings indicate that increasing TCP1 expression will increase the cell growth and the sensitivity to FTY720, further confirming that FTY720 can target TCP1 and inhibit AML cell proliferation.

### 3.7. FTY720 Could Induce Apoptosis and Cell Cycle Arrest in AML Cells

Because TCP1 plays a vital role in cellular biological processes, we detected the effect of FTY720 on apoptosis induction and cell cycle arrest. As shown in Figure 6A,B the early and late stages of the apoptosis rate of the three AML cell lines increased rapidly as FTY720 increased, and HL-60-G4 cells also showed more sensitivity to FTY720 than HL-60 cells. We also found that after treatment with FTY720, caspase family-related proteins changed significantly, which meant that FTY720 induced the apoptosis of AML cells by activating the caspase family (Figure 6C–E). As shown in Figure 7A,B HL-60-G4 cells treated with FTY720 showed cell cycle arrest in both the G0/G1 phase and G2/M phase, and HL-60-G4 cells treated with 15 μmol·L^−1^ FTY720 showed severe G2/M arrest. These findings indicated that FTY720 might arrest the cell cycle in the G1 phase by binding with TCP1 to inhibit its function, and FTY720 also influenced some other processes to induce cell cycle arrest in the G2 phase.

### 3.8. FTY720 Decreased Protein Levels by Inhibiting TCP1 Chaperone Function

It was reported that TCP1 could affect the PLK1 active form to influence cell cycle transition, and TCP1 itself is also involved in cell cycle regulation. We also found that the protein levels of PLK1, CDC20, MYC, and PKMYT1 in AML cells treated with increasing concentrations of FTY720 for 24 h were significantly decreased (Figure 7C,D). As the cell cycle-related proteins decreased, ubiquitin levels in AML cells increased. Then, we speculated that PLK1 was the potential client protein of TCP1 and used coimmunoprecipitation (Co-IP) to further clarify whether the reduction in PLK1 protein occurred through inhibiting the chaperone function of TCP1. After treatment with 30 μmol L^−1^ FTY720 for 24 h, the interaction between TCP1 and PLK1 in HL-60-G4 cells was disrupted, confirming that FTY720 could inhibit the function of TCP1 to decrease the intracellular protein level of PLK1 (Figure 7E). Moreover, it was reported that CDC20, which is related to the cell cycle, was also the client protein of TCP1. Co-IP results show that FTY720 could influence the interaction between TCP1 and CDC20 (Figure 7F). Interestingly, MYC and PKMYT1 are substrates of PLK1 and CDC20, and FTY720 may affect the binding of TCP1 to PLK1 and CDC20 to further influence the levels of MYC and PKMYT1.

Furthermore, treating Kasumi-1 cells for 24 h with FTY720 could decrease the protein level of AML1-ETO (Figure 7G), a fusion oncoprotein, which is the most common cytogenetic subtype of AML. We also verified that TCP1 could interact with AML1-ETO, which was disrupted by 40 μmol L^−1^ FTY720 treatment for 24 h (Figure 7H). These findings indicate that FTY720 could reduce the protein levels of PLK1, CDC20, MYC, PKMYT1, and AML1-ETO by interfering with the function of TCP1. FTY720 could even have a specific therapeutic effect on AML with AML1-ETO.

### 3.9. FTY720 Inhibited the Growth of Xenograft AML Tumors in Vivo and Extended the Survival Time of Mice

The tumor inhibition rate in the 5 mg·kg^−1^ FTY720 group was 61.99%, and this rate reached 72.73% in the 10 mg·kg^−1^ FTY720 group, which meant that FTY720 could significantly inhibit the growth of tumors (Figure 8A). It is worth noting that the body weight of mice in the 10 mg·kg^−1^ FTY720 group decreased slightly compared to the control group and remained unchanged in the 5 mg·kg^−1^ FTY720 group, which meant that FTY720 had few severe adverse reactions in the living mice (Figure 8B). Furthermore, we also found that the 10 mg·kg^−1^ FTY720 group exhibited longer survival than the 5 mg·kg^−1^ FTY720 group (Figure 8C). It was interesting that the tumor of a mouse in the 10 mg·kg^−1^ FTY720 group turned black and dry, and there was no tumor in that mouse, which also had the longest survival time. These findings indicated that FTY720 could significantly slow tumor development without serious adverse effects and might have unexpected antitumor effects.

## 4. Discussion

Since 2017, the FDA has approved several clinical treatment regimens for AML, including small molecule inhibitors, such as FLT3 inhibitors, antibody-drug conjugates, and cytotoxic agents. The clinical combination of these new drugs provides a new treatment option for AML patients [3,29]. For patients with different clinical, cytogenetic, and mutational characteristics, it is clinically necessary to evaluate which treatment options are suitable for patients to achieve the highest benefit. Although new therapeutic targets and drugs continue to emerge, prolonging the survival period, the patients ideal for these treatments are relatively limited, and the vast majority of patients are facing incomplete cures or relapse [27]. Therefore, it is still necessary to continuously study new targets and drugs for the clinical treatment of AML.

In our study, TCP1 was found to be an effective treatment target for AML. In the mouse AML model, it was found that the expression of TCP1 increased with the increasing tumor generation. Our group previously found that TCP1 can increase the drug resistance of AML by activating the AKT/mTOR signaling pathway [4]. Here, the protein profile of TCP1 knockdown in AML cells showed that the affected signaling pathway also contained the PI3K-AKT pathway. In addition, it also affects oxidative phosphorylation, cellular senescence, and Th17 cell differentiation. We also found that the knockdown of TCP1 could influence the proliferation of AML cells in vivo and in vitro. These indicate that targeting TCP1 can indirectly affect the above biological processes, thereby affecting the growth and proliferation of AML. Therefore, we further studied the drugs and miRNAs that can target and inhibit TCP1.

We found that miR-340-5p was downregulated in AML patients, and the same finding was shown in the work of Yuan Liu [30]. Many researchers found that miR-340-5p was significantly downregulated and involved in promoting tumor growth in diffuse large B-cell lymphoma [31], colon cancer [32], non-small cell lung cancer [33], glioblastoma multiforme [34], etc., which was consistent with our results. Our results also found for the first time that miR-340-5p can specifically bind to TCP1 mRNA to inhibit the expression of TCP1, and TCP1 was proven to be the direct target gene of miR-340-5p. This result showed that miR-340-5p could inhibit the proliferation of AML cells by targeting the TCP1.

We also confirmed that FTY720 can specifically bind to TCP1 through virtual screening and Biacore experiments and affect the physiological function of AML cells. Many studies have found that FTY720, as a PP2A activator, has efficient killing and growth inhibitory effects on AML cells [23,35]. In this study, we found that FTY720 can also inhibit the function of TCP1 by disrupting the interaction between TCP1 and client proteins, inducing cell cycle arrest and apoptosis to inhibit and kill AML cells. However, considering the toxicity caused by the activation of the S1P receptor of FTY720 [36], FTY720 cannot be directly used in AML patients, and structural modification is needed to reduce its adverse effects of immunosuppression to achieve better anti-AML outcomes with lower toxicity.

## 5. Conclusions

Our study delineates TCP1 as a master regulator of AML progression and a therapeutically actionable target. Mechanistically, elevated TCP1 drives AML tumorigenesis by stabilizing oncoproteins (AML1-ETO/PLK1), while its suppression triggers proteasomal degradation of these targets. Therapeutically, we validated two strategies: miR-340-5p-mediated transcriptional silencing and FTY720-induced functional inhibition of TCP1 chaperone activity. Both approaches achieved robust anti-leukemic effects in vitro and in vivo, including proliferation arrest, apoptosis induction, and tumor regression. This study provides a unique research basis for the exploration of the mechanism of AML. Future studies should focus on translating these preclinical insights into clinical applications, including optimizing delivery strategies for miR-340-5p and evaluating FTY720 in combinatorial regimens to overcome AML heterogeneity and resistance.

## Figures and Tables

**Figure 1 pharmaceutics-17-00557-f001:**
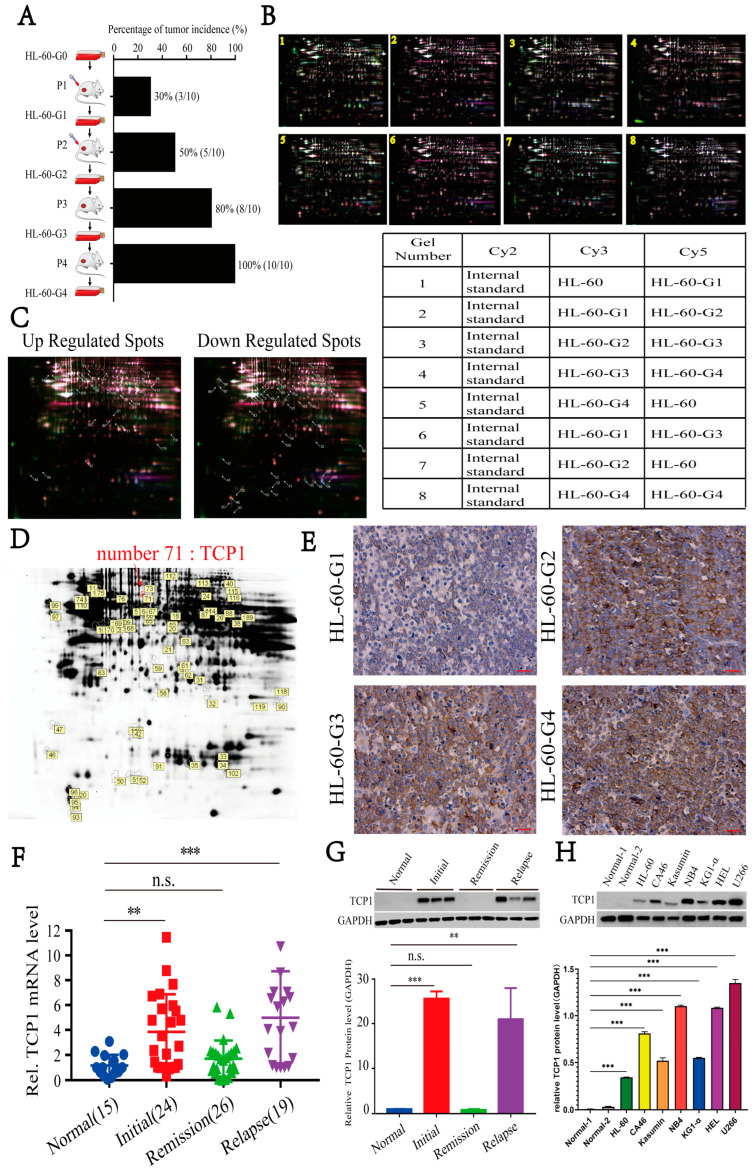
TCP1 is positively related to the tumorigenesis and development of AML. (**A**) In the successive generations, the tumor formation rate increased in the HL-60 cell line. (**B**) Mass spectrometry combined with two-dimensional in-gel electrophoresis revealed that TCP1 is a highly oncogenic gene. (**C**) Differentially expressed protein spots marked with master numbers displayed in 2D-DIGE images. The arrows indicate the 33 significantly upregulated protein spots (marked with master numbers) and 54 downregulated protein spots (marked with master numbers) displayed in 2D-DIGE images. (**D**) Proteomic analysis of HL-60 cells using 2D-DIGE. The arrows indicate protein spots that differed significantly between the HL-60 cells and the control cells. Relative spot intensity was calculated on the basis of the spot volume. Only the spots indicated with arrows were used for identification. (**E**) IHC staining examined TCP1 expression in tumor tissues of different generations. The TCP1 expression in tissue samples of different stage of AML patients was detected by RT-qPCR (**F**) and Western blotting (**G**). (**H**) Western blotting was used to examine the expression of TCP1 in different leukemia cell lines. Data are presented as the means ± SDs from three independent experiments. ** *p* < 0.01, *** *p* < 0.001, ns: no significance.

**Figure 2 pharmaceutics-17-00557-f002:**
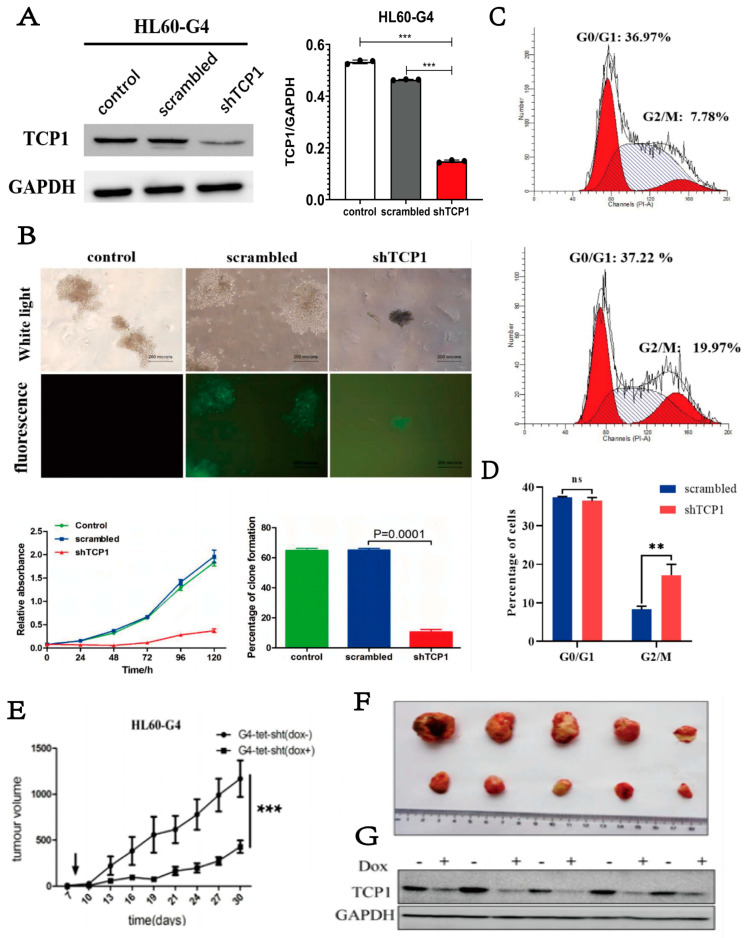
The effect of TCP1 on AML cell development and proliferation in vitro and in vivo. (**A**) The efficiency of shTCP1 was measured in HL-60-G4 cells by Western blotting. (**B**) A clonogenic cell survival assay was performed to evaluate the growth of HL-60-G4-shTCP1 cells in vitro. (**C**,**D**) Representative cell cycle distribution of HL-60-G4 cells after knocking down TCP1 with lentivirus transfection. (**E**) Detection of xenograft tumor formation in nude mice using Tet-on technology. (**F**) Representative image of Dox- and Dox+ groups of tumors at the end of the experiment. (**G**) Western blotting was used to analyze the protein level of TCP1 in tumor tissues in each group. Data are presented as the means ± SD from three independent experiments. ** *p* < 0.01, *** *p* < 0.001, ns: no significance.

**Figure 3 pharmaceutics-17-00557-f003:**
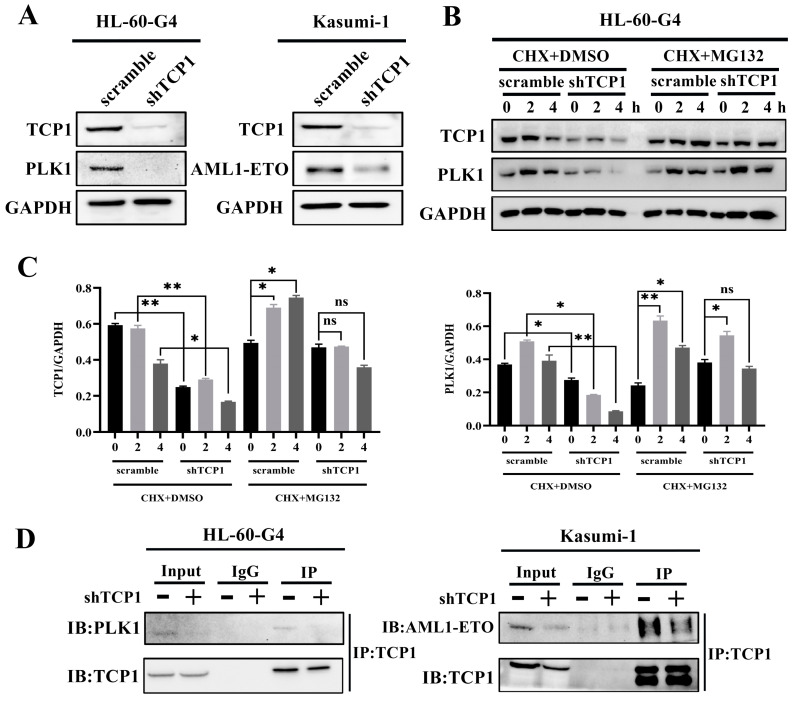
TCP1 regulated the stability of PLK1 and AML1-ETO. (**A**) Western blotting analysis of TCP1, PLK1, and AML1-ETO expression in HL-60-G4-shTCP1 and Kasumi-1-shTCP1 cells. (**B**) Cells were incubated with 10 μM CHX alone or combined with 10 μM MG132, and WB assays were used to determine the levels for TCP1 and PLK1. (**C**) The image shown in Figure 3B was quantified using ImageJ software (Version 1.8.0). (**D**) Co-IP assays were performed to observe the pull-down levels of PLK1 and AML1-ETO in the shTCP1 group and scramble group. Data are presented as the means ± SD from three independent experiments. * *p* < 0.05, ** *p* < 0.01, ns: no significance.

**Figure 4 pharmaceutics-17-00557-f004:**
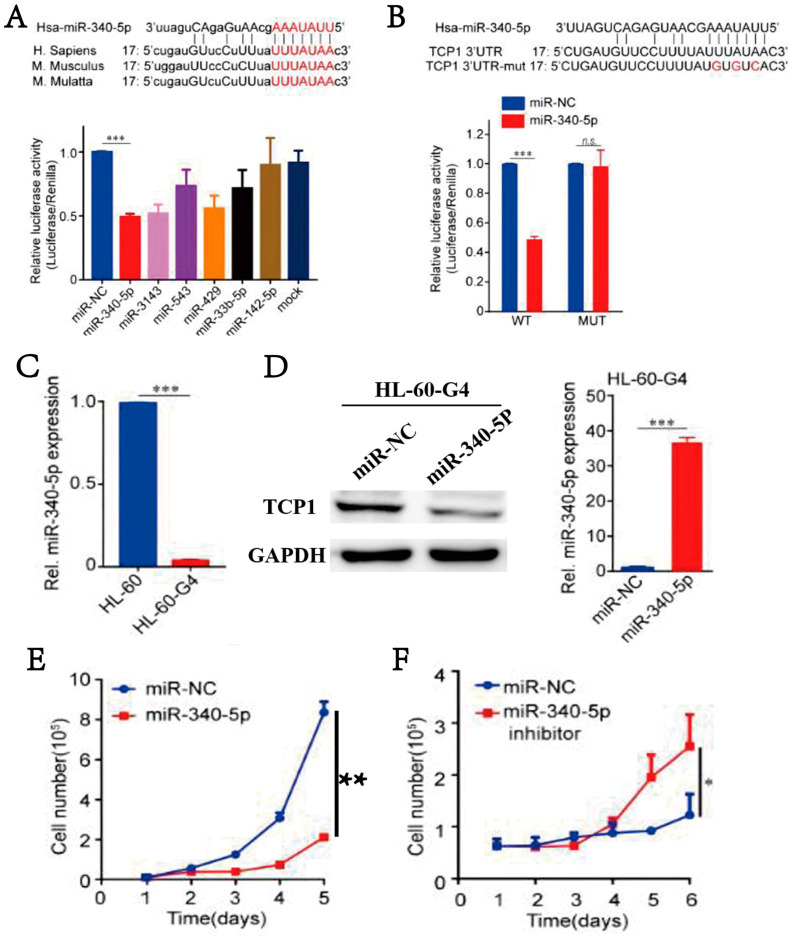
MiR-340-5P inhibits AML cell proliferation by silencing TCP1. (**A**) Dual-luciferase reporter system analysis of targeted transcriptional regulation of TCP1 by miRNAs. (**B**) HEK293 cells were co-transfected with miR-340-5p mimics or control and reporter plasmid or the mutant 3′-UTR of TCP1, and the luciferase activity was measured after 48 h. (**C**) MiR-340-5p levels in AML cell lines were tested by RT-qPCR. (**D**) HL-60-G4 cells overexpressing (O/E) the vector control or miR-340-5p. Detection of TCP1 expression in HL-60-G4 cells overexpressing miR-340-5p by Western blot. (**E**) The cellular proliferation of HL-60-G4^O/E^ cells was determined by cell counting. (**F**) HL-60-G4^O/E^ cells were treated with 100 nmol L^−1^ of an hsa-miR-340-5p inhibitor for 24 h, and cellular proliferation was examined. Data are presented as the means ± SDs from three independent experiments. * *p* < 0.05, ** *p* < 0.01, *** *p* < 0.001, ns: no significance.

**Figure 5 pharmaceutics-17-00557-f005:**
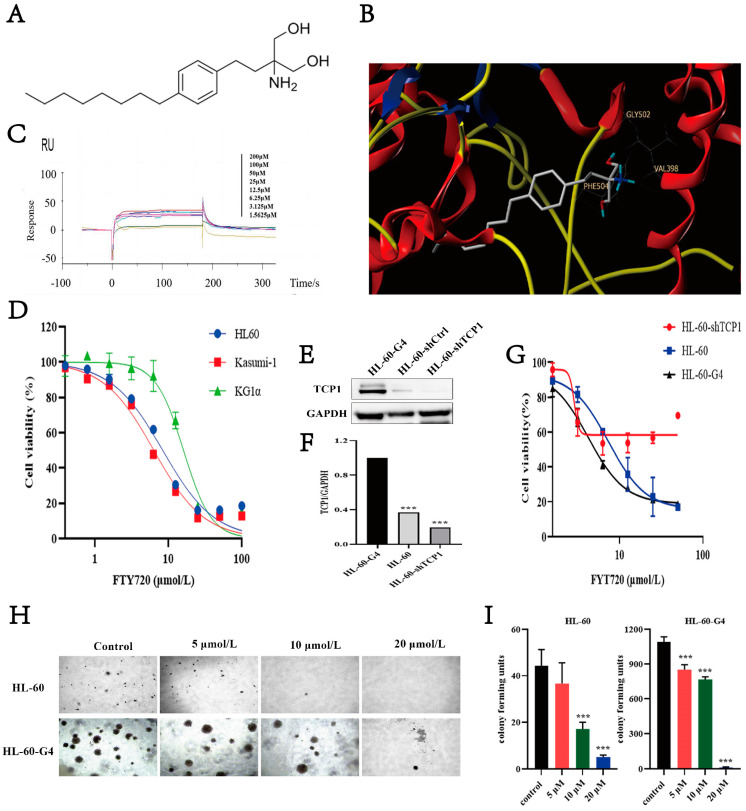
FTY720 has an affinity for TCP1 and can inhibit AML cell proliferation. (**A**) The chemical structure of FTY720. (**B**) TCP1 subunit protein and FTY720 molecular docking virtual screening. (**C**) Surface plasmon resonance detected binding between FTY720 and TCP1. (**D**) The effect of FTY720 on the viability of AML cells. (**E**) TCP1 protein levels in the three cell lines were examined by Western blotting and (**F**) normalized to the corresponding density of GAPDH. (**G**) The sensitivity of HL-60 cells to FTY720 increased as the TCP1 protein level increased. (**H**) HL-60-G4 cells formed many more colony units than HL-60 cells, and their colony numbers decreased as FTY720 increased. (**I**) The quantification and plot were created using ImageJ software (Version 1.8.0) for the image shown in Figure 5H. Data are presented as the means ± SDs from three independent experiments. *** *p* < 0.001.

**Figure 6 pharmaceutics-17-00557-f006:**
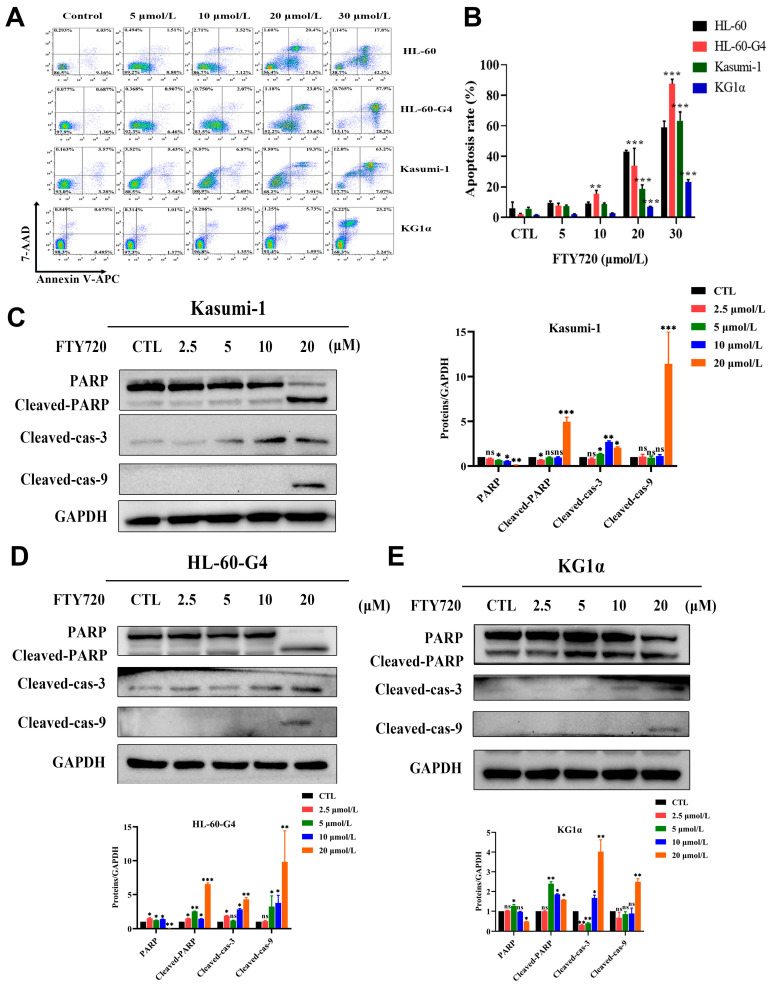
Induction of apoptosis by FTY720 in AML cells. (**A**) Representative flow cytometry images. (**B**) Quantitative analysis of the percentage of apoptotic cells after treatment with FTY720 for 24 h. (**C**–**E**) Western blotting analysis of caspase family-related proteins in AML cells exposed to different concentrations of FTY720. Data are presented as the means ± SDs from three independent experiments. * *p* < 0.05, ** *p* < 0.01, *** *p* < 0.001, ns: no significance.

**Figure 7 pharmaceutics-17-00557-f007:**
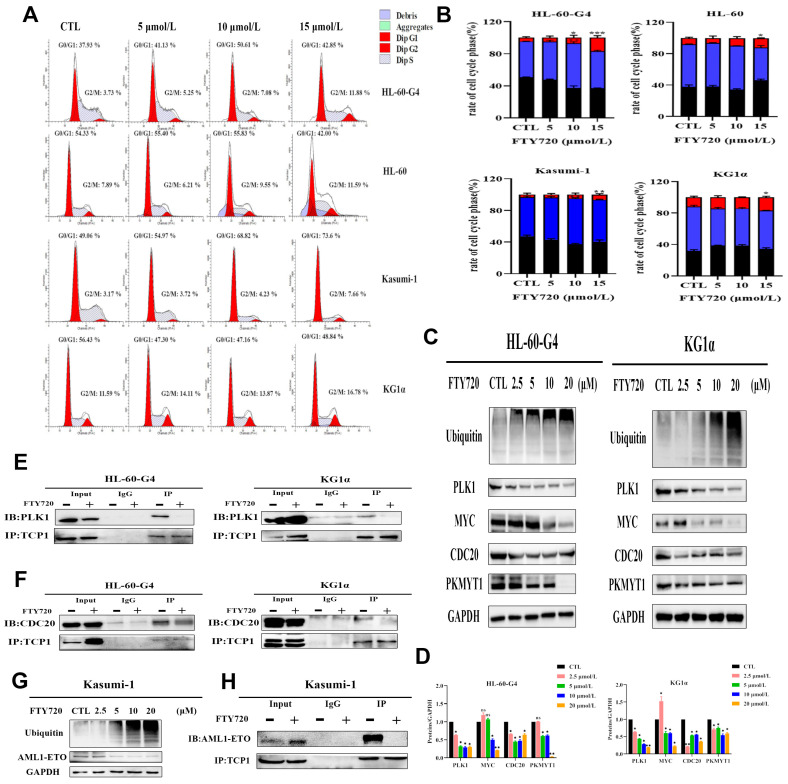
FTY720 induces cell cycle arrest and decreases protein levels by inhibiting TCP1 chaperone function in AML cells. (**A**) Representative cell cycle distribution images. (**B**) Column chart of the cell cycle ratio of HL-60-G4, HL-60, Kasumi-1, and KG1α cells. (**C**) Changes in intracellular protein levels in HL-60-G4 and KG1α cells incubated with FTY720 for 24 h. (**D**) The image shown in Figure 8C was quantified using ImageJ software (Version 1.8.0). (**E**) The interaction between the PLK1 protein and TCP1 protein in HL-60-G4 and KG1α cells exposed to FTY720 for 24 h was assayed by Co-IP. (**F**) The interaction between the CDC20 protein and TCP1 protein in HL-60-G4 and KG1α cells exposed to FTY720 for 24 h was assayed by Co-IP. (**G**) Changes in intracellular protein levels in Kasumi-1 cells incubated with FTY720 for 24 h were detected by Western blotting. (**H**) TCP1 interacts with AML1-ETO in Kasumi-1 cells, but FTY720 disrupts the interaction. Means ± SDs of three independent experiments are shown. * *p* < 0.05, ** *p* < 0.01, *** *p* < 0.001, ns: no significance.

**Figure 8 pharmaceutics-17-00557-f008:**
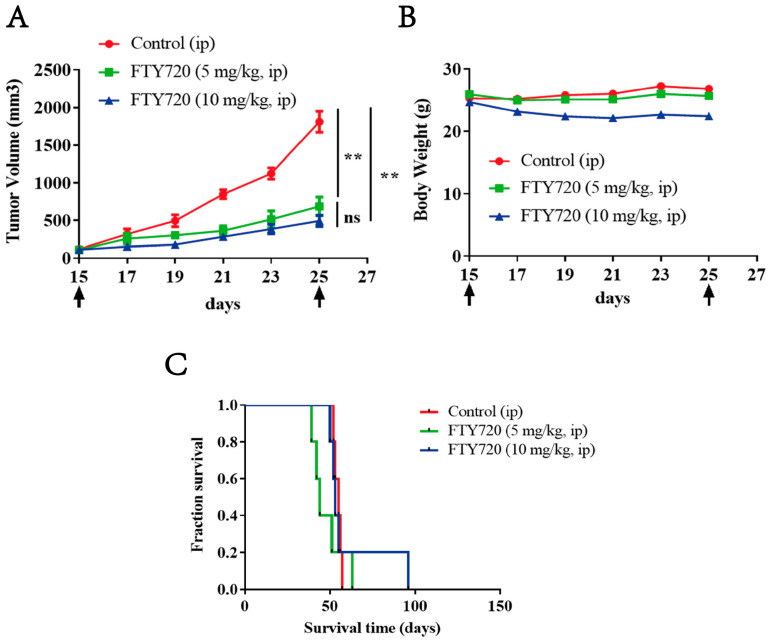
Anti-tumor effect of FTY720 on xenograft AML tumors. (**A**) The tumor volume in each group, *n* = 5. (**B**) Weight of tumor-bearing mice during the treatment period. (**C**) Survival curves of mice treated with FTY720. ** *p* < 0.01, ns: no significance.

## Data Availability

The data that support the findings of this study are available from the corresponding author upon reasonable request. Some data may not be made available because of privacy or ethical restrictions.

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
