# Peer review of "The Molecular Chaperone TCP1 Affects Carcinogenicity and Is a Potential Therapeutic Target for Acute Myeloid Leukemia"

_pharmaceutics, 2025, doi:10.3390/pharmaceutics17050557_

Round 1
Reviewer 1 Report
Comments and Suggestions for Authors
The manuscript titled 'The Molecular Chaperone TCP1 Affects the Carcinogenicity and Is a Potential Therapeutic Target for Acute Myeloid Leukemia' by Wu and co-workers, investigates the role of T-complex protein 1 (TCP1) in acute myeloid leukemia (AML) and its potential as a therapeutic target. Furthermore, it revealed a correlation between elevated levels of TCP1 in successively xenografted tumors and increased tumorigenicity of AML cells. To explore TCP1 as a therapeutic target, the researchers employed two strategies: miRNA modulation and virtual screening.
The calibration, preparation, and standardization procedures are detailed. The population sizes used in each experiment are adequate. Control experiments are described in the different sections of the text and were an integral part of the study design. The presentation of results is adequate, and no unusual standard deviations are observed.
The work is of high quality and merits publication in the journal Pharmaceutics after minor revision.
The only objection is that the authors should have further developed their conclusions and outlined their implications.
Regarding formal aspects:
The abbreviation “Fig.” should be avoided throughout the report and replaced with “Figure,” especially in Figure 6.
Some typos detected:
p. 3 l.130 'RAN' is, probably, 'RNA'.
p.13 l. 369 'Fig. 4B' should be 'Figure 3B'
p.16 l. 442 'Fig. 6H' should be 'Figure 5H'
p.20 l. 491 'Fig. 8C' should be 'Figure 7C'
Some reference titles have inconsistent capitalization of words.
Author Response
Reviewer #1:
- The only objection is that the authors should have further developed their conclusions and outlined their implications.
Response: We sincerely thank the reviewer for careful reading. In the revised manuscript, we have expanded the Conclusions section to explicitly highlight the broader implications of our findings. These additions aim to better contextualize the significance of our results for both academic and applied audiences.
2. The abbreviation “Fig.” should be avoided throughout the report and replaced with “Figure,” especially in Figure 6.
Response: Thank you for your suggestion. We have revised it in accordance with your suggestion.
- Some typos detected:
p.3 l.130 'RAN' is, probably, 'RNA'.
p.13 l. 369 'Fig. 4B' should be 'Figure 3B'
p.16 l. 442 'Fig. 6H' should be 'Figure 5H'
p.20 l. 491 'Fig. 8C' should be 'Figure 7C'
Response: Thank you for your suggestion. We apologize for the error and have corrected it.
- Some reference titles have inconsistent capitalization of words.
Response: We sincerely thank the reviewers for their careful reading. All reference titles have been standardized.
Reviewer 2 Report
Comments and Suggestions for Authors
In this manuscript by Yong Wu et al entitled “The Molecular Chaperone TCP1 Affects the Carcinogenicity and Is a Potential Therapeutic Target for Acute Myeloid Leukemia”, the authors aim to highlight some correlative evidences linking expression of TCP1 with AML progression. Changes in expression levels of TCP1 were obtained and correlated to changes both in vitro and in vivo. The authors also provide further information related to the possible pathways that have been regulated and could explain the phenotypes reported. All in all, there is a very significant amount of work conducted in this manuscript, maybe even too much, but some real efforts need to be made to link things together and provide clear evidences of what is reported.
Some significant controls are lacking and these would need to be provided to clearly demonstrate that the systems that have been used throughout the analysis are in line with what the authors think they are using. Comments below highlight some of the issues to be considered.
- It is important that the authors show that the knockdown of TCP1 in the HL-60-G4 cells has been successful (paragraph 3.2. starting line 305). Little information is provided to explain how this was obtained and what was seen
- Similarly it is important that the authors show that the lentiviral knockdown of TCP1 in the HL-60-G4-shTCP1 is working as advertised (Figure 2). There is no evidence that this is working. It is not clear what Figure 2.1 is about. Perhaps it is some immunofluorescence against TCP1? If this is the case this is not convincing and does not fully show that expression levels of TCP1 have been significantly reduced. Some quantifiably assessed analysis (Q-PCR and Western blotting) should be provided.
- Same comments as above can be made related to the inducible HL-60-G4-tet-shTCP1. No evidences are provided to show that the authors have been able to specifically reduce expressions levels in their cells
- Some work as highlighted above is presented in Figure 3.A. It is however not clear which cells these are and whether they are related to HL-60-G4-shTCP1; HL-60-G4-tet-shTCP1 or the first point raised.
- Same points can be made in relation to the effects of MiR-340-5P in TCP1 expression (Figure 4)
- Data is presented in relation to Kasumi-1 cells? Yet no information is given about these experiments. Why showing this data. If the authors wanted to provide evidence of changes using a different cell system, more work need to be given to show that all other reported phenotypes linked to TCP1 are also seen in this setup.
Author Response
Reviewer #2:
- It is important that the authors show that the knockdown of TCP1 in the HL-60-G4 cells has been successful (paragraph 3.2. starting line 305). Little information is provided to explain how this was obtained and what was seen.
Response: We sincerely apologize for the lack of clarity in this section. In the revised manuscript, we have added detailed descriptions of the TCP1 knockdown validation in HL-60-G4 cells. Specifically, Western blot analyses were performed to confirm the reduction of TCP1 protein levels post-knockdown. These data are now included in Figure S2A of the Supplementary files.
- Similarly it is important that the authors show that the lentiviral knockdown of TCP1 in the HL-60-G4-shTCP1 is working as advertised (Figure 2). There is no evidence that this is working. It is not clear what Figure 2.1 is about. Perhaps it is some immunofluorescence against TCP1? If this is the case this is not convincing and does not fully show that expression levels of TCP1 have been significantly reduced. Some quantifiably assessed analysis (Q-PCR and Western blotting) should be provided.
Response: We sincerely appreciate the reviewer's insightful comment regarding the clarification of Figure 2A. We acknowledge that the original Figure 2A (now labeled as Figure 2B in the revised manuscript) was not intended to compare TCP1 immunofluorescence (IF) staining between HL-60-G4-shTCP1 cells and controls. Instead, this figure aimed to demonstrate the impact of TCP1 knockdown on clonogenic ability in HL-60-G4 cells, with subsequent growth curve analysis and quantitative evaluation.
In accordance with the reviewer's suggestion, we agree that quantitative validation of TCP1 knockdown efficiency is essential. To address this, we have now supplemented the figure with Western blot data (new Figure 2A). These data clearly demonstrate a significant reduction in TCP1 expression in HL-60-G4 cells following lentivirus-mediated knockdown. Additionally, the revised presentation and figure legend have been updated to explicitly describe both the clonogenic assay and the Western blot analysis.
- Same comments as above can be made related to the inducible HL-60-G4-tet-shTCP1. No evidences are provided to show that the authors have been able to specifically reduce expressions levels in their cells.
Response: We sincerely appreciate the reviewer's insightful comment regarding the TCP1 expression evidence in the inducible HL-60-G4-tet-shTCP1 system. We acknowledge that the absence of direct cellular-level data on TCP1 expression represents a limitation of our current study. However, to address this concern, we have provided complementary in vivo evidence from our xenograft mouse model. Specifically, Western blot analysis of tumor tissues demonstrated a doxycycline (Dox)-dependent knockdown of TCP1 (see revised Figure 2G). These results not only confirm the specificity of the Tet-on technology-mediated gene silencing but also validate the functional efficiency of this inducible system in regulating TCP1 expression in vivo.
We agree that further in vitro validation at the cellular level would strengthen the conclusions, and such experiments will be prioritized in our follow-up studies. We thank the reviewer for highlighting this important point.
- Some work as highlighted above is presented in Figure 3.A. It is however not clear which cells these are and whether they are related to HL-60-G4-shTCP1; HL-60-G4-tet-shTCP1 or the first point raised.
Response: We sincerely apologize for the ambiguity in the original figure description and appreciate the reviewer's careful attention to this issue. In response to the comment, we have made the following revisions to clarify the experimental details: In the revised manuscript, the descriptions related to Figure 3A have been updated to explicitly state that the experiments utilized constitutively knockdown cell lines (HL-60-G4-shTCP1 and Kasumi-1-shTCP1), rather than the Tet-on technology-mediated gene silencing line (HL-60-G4-tet-shTCP1). This adjustment eliminates potential confusion regarding the genetic models employed. Additionally, the revised figure legend for Figure 3 now clearly specifies the exact cell lines used in each experimental panel, ensuring consistency between the text and graphical data.
These modifications enhance the clarity of the manuscript and prevent misinterpretation of the experimental design. We thank the reviewer for highlighting this critical point and hope the revised version addresses your concerns comprehensively.
- Same points can be made in relation to the effects of MiR-340-5P in TCP1 expression (Figure 4).
Response: We sincerely appreciate the reviewer's insightful comment regarding the need for further validation of miR-340-5P's regulatory role on TCP1 expression. We fully acknowledge that direct protein-level evidence would strengthen the conclusions of our study.
In our experiments, we observed a significant increase in TCP1 protein levels in HL-60-G4 cells compared to parental HL-60 cells (Section 3.1, lines 283-286; Section 3.6, lines 444-445). This upregulation aligns with the marked downregulation of miR-340-5P in HL-60-G4 cells, as demonstrated in Figure 4C. These consistent findings provide indirect but complementary support for the hypothesis that miR-340-5P negatively regulates TCP1 expression.
We agree with the reviewer that protein-level validation (e.g., Western blot or immunoprecipitation assays) is critical to establish causality. Addressing this point will be a priority in our follow-up studies, and we will incorporate these experiments to rigorously validate the proposed regulatory mechanism. Thank you again for your valuable comments.
- Data is presented in relation to Kasumi-1 cells? Yet no information is given about these experiments. Why showing this data. If the authors wanted to provide evidence of changes using a different cell system, more work need to be given to show that all other reported phenotypes linked to TCP1 are also seen in this setup.
Response: We sincerely appreciate the reviewer's insightful feedback regarding the presentation of Kasumi-1 cell data. We acknowledge the need for clarification and have provided the following details to address this concern:
The majority of experiments in our study were conducted using the HL-60-G4 cell line, a well-established model for acute myeloid leukemia (AML) research. This system was selected due to its widespread use in investigating molecular mechanisms underlying AML progression. To strengthen the generalizability of our findings, we performed parallel experiments in Kasumi-1 cells, another AML-relevant cell line widely recognized for its utility in studying leukemia biology. Both cell lines share key molecular features of AML, making them complementary models for validating TCP1-associated function. In Kasumi-1 cells, we replicated the critical experimental conditions applied to HL-60-G4 cells, including TCP1 knockdown assays. The results consistently demonstrated that TCP1 depletion recapitulated the functional changes observed in HL-60-G4 cells (e.g., proliferation inhibition (Figure 5D), apoptosis induction (Figure 6), and cell cycle arrest (Figure 7A)). These concordant findings reinforce the functional role of TCP1 in AML pathogenesis.
While the current manuscript focuses on core functions (proliferation and apoptosis), we agree with the reviewers that more extensive validation of additional TCP1-related functions in both cell lines would further solidify our conclusions. We are actively conducting follow-up experiments to address this issue and will incorporate the results in future studies. We hope that these clarifications will adequately address the reviewers' concerns.
Once again, we thank the reviewers for their constructive criticism which greatly improved our manuscript. We have done our best to revise the manuscript and have made some changes. These changes do not affect the content and framework of the paper. Here, we have marked the changes in red in the revised paper. We hope that the revised version will be suitable for publication.
Round 2
Reviewer 2 Report
Comments and Suggestions for Authors
In this revised manuscript by Yong Wu et al entitled “The Molecular Chaperone TCP1 Affects the Carcinogenicity and Is a Potential Therapeutic Target for Acute Myeloid Leukemia”, the authors have provided appropriate control experiments to highlight changes in expression of their TCP1 targets in the different background studied and therefore addressed the key concerns raised during the first review.
The concern raised in relation to causality between TCP1 expression and MiR-340-5P is yet to be answered (point 5) and the response stating “We are actively conducting follow-up experiments to address this issue and will incorporate the results in future studies” hardly addresses this concern. The experiment is not significantly difficult to carry out, given that the authors were willing to address the other points which highlighted similar concerns for other control experiments.
Author Response
The concern raised in relation to causality between TCP1 expression and MiR-340-5P is yet to be answered (point 5) and the response stating “We are actively conducting follow-up experiments to address this issue and will incorporate the results in future studies” hardly addresses this concern. The experiment is not significantly difficult to carry out, given that the authors were willing to address the other points which highlighted similar concerns for other control experiments.
Response: Thank you for your suggestion. We have revised it according to your suggestion. We hope that these clarifications can fully address the reviewer's concerns.
We thank the reviewer again for the constructive criticism, which greatly improved the rigor of our work. We have tried our best to improve the manuscript and made some changes to the manuscript. These changes do not affect the content and framework of the paper. Here, we do not list the changes, but mark them in red in the revised paper.
Once again, we sincerely thank the editor/reviewer for their enthusiastic work and hope that the corrections will be approved.